# Role of Hydrogen Peroxide Vapor (HPV) for the Disinfection of Hospital Surfaces Contaminated by Multiresistant Bacteria

**DOI:** 10.3390/pathogens9050408

**Published:** 2020-05-24

**Authors:** Michele Totaro, Beatrice Casini, Sara Profeti, Benedetta Tuvo, Gaetano Privitera, Angelo Baggiani

**Affiliations:** Department of Translational Research and the New Technologies in Medicine and Surgery, University of Pisa, 56123 Pisa, Italy; micheleto@hotmail.it (M.T.); beatrice.casini@med.unipi.it (B.C.); profeti.sara@gmail.com (S.P.); tuvobenedetta@hotmail.it (B.T.); gaetano.privitera@med.unipi.it (G.P.)

**Keywords:** hydrogen peroxide vapor, multidrug-resistant bacteria, hospital disinfection

## Abstract

The emergence of multiresistant bacterial strains as agents of healthcare-related infection in hospitals has prompted a review of the control techniques, with an added emphasis on preventive measures, namely good clinical practices, antimicrobial stewardship, and appropriate environmental cleaning. The latter item is about the choice of an appropriate disinfectant as a critical role due to the difficulties often encountered in obtaining a complete eradication of environmental contaminations and reservoirs of pathogens. The present review is focused on the effectiveness of hydrogen peroxide vapor, among the new environmental disinfectants that have been adopted. The method is based on a critical review of the available literature on this topic

## 1. Introduction

The disinfection of hospital surfaces is a complex operation aimed at reducing the pathogenic microorganism load. An ideal disinfectant must be safe for human health. It may have a good stability in the environment and may be free of toxic activity [1,2,3,4]. 

Hydrogen peroxide is a versatile disinfectant, since it can be used in several environmental matrices: air, water, wastewater, surfaces, soil, etc. It may be used in combination with other agents increasing the disinfection times. Hydrogen peroxide is more oxidizing than chlorine and chlorine dioxide. The disinfection mechanism is based on the release of oxygen free radicals, which cause genomic damage in bacterial cells [5,6].

Hydrogen peroxide in the vaporized form (HPV) is used for surface disinfection. It may be combined with higher or lower concentrations of silver ions [7,8,9].

HPV bactericidal activity may be experimentally evaluated in vitro [10] or in hospital settings [11], in order to provide scientific data related to:the type of microbial strains sensitive or resistant to the compound;the adequate chemical concentrations; andthe contact time.

The management of the spread of multidrug-resistant bacteria (MDR) in the hospital setting [12,13] is a crucial issue that needs the evaluation of HPV activity.

The incidence of infectious outbreaks from antibiotic resistant microorganisms is becoming one of the main problems in hospital contexts [14]. Considering the extreme difficulty in MDR eradication, preventive measures, including an accurate environmental disinfection, are becoming increasingly necessary [15,16,17,18,19].

All hospital settings need an accurate evaluation of new disinfection efficacy against bacterial species such as carbapenem-resistant enterobacteria, multidrug-resistant *Acinetobacter baumannii,* and spore-forming bacteria (*Clostridium difficile),* which are frequently responsible for nosocomial infections [20].

The aim of this review is to evaluate the main studies performed on HPV activity for MDR prevention in vitro and in hospital settings [21].

## 2. In Vitro Experimental Test Performed with Hydrogen Peroxide (HP) and Hydrogen Peroxide Vapor (HPV)

The bactericidal activity of hydrogen peroxide was studied in several in vitro tests. In 1987, a Californian working group published data related to the ability of hydrogen peroxide (30–100 ppm) to cause DNA damage in different *Escherichia coli* strains, mostly in *oxy*R and Son of Sevenless (SOS) regions, involved in the shelter of genomic damage caused by oxidizing agents [22,23].

Further in vitro studies were performed by the Asian group of Absalan [24], who demonstrated the HP bactericidal activity on several strains of enterobacteria, such as *E. coli*, *Proteus mirabilis,* and *Klebsiella pneumoniae*. The antimicrobial activity of the disinfectant was proved in bacterial suspension and on contaminated steel surfaces (20 cm^2^), applying 0.3% HPV with 30 ppb silver ions. The microbial growth was monitored at different concentration ranges and for a total time of 24 h. A significant reduction in microbial loads was detected on surfaces (*p* = 0.008 for *E. coli*, *p* = 0.014 for *Klebsiella pneumoniae* and *p* = 0.002 for *Proteus mirabilis*).

An Italian research group [25] tested in vitro the activity of hydrogen peroxide with silver ions in colonized surfaces. The study evaluated the biocidal effectiveness on *Staphylococcus aureus* American Type Culture Collection (ATCC) 6538 strain, *Pseudomonas aeruginosa* ATCC 15442 strain, and several clinical isolates of multidrug-resistant *Staphylococcus aureus* and *Pseudomonas aeruginosa.* A good disinfectant activity was obtained in suspension and on surfaces. The efficacy of the biocidal compound varied according to the different action times.

Further studies [26,27] evaluated the HPV disinfection activity under similar conditions to those of the hospital environment, by using experimental technologies for environmental decontamination through aerosolization.

Lemmen [26] assessed the effectiveness of an HPV disinfection device on two different types of surfaces, stainless steel and cotton, highlighting any differences in susceptibility to bacteria placed on porous supports (cotton). *Clostridium difficile* spores and three antibiotic-resistant strains, MDR *Acinetobacter baumannii*, vancomycin-resistant (VRE) *Enterococcus faecalis*, and methicillin-resistant *Staphylococcus aureus* (MRSA) were selected for the study. The disinfection procedure was divided into three HPV nebulization cycles, for a total of 50–52 min. The HPV concentration peak measured in the air was 500–600 ppm, followed by room aeration, for a total of 2–3 h. Results showed a significant reduction of all microbial species between 4.0 log (VRE) to 5.1 log (MDR *A. baumannii*). Differences between the different positions and supports (steel and cotton) were not observed. This study demonstrated the effectiveness of HPV on porous surfaces also, a result that was not always guaranteed by disinfectants [28].

A similar work was carried out by Herruzo [27], who tested an HPV aerosolization machine in a room contaminated by carriers in order to simulate a dirty condition. Carriers were placed at different distances from the device, between 0 and 6 m, while controls were placed in an adjacent room not exposed to the HPV activity. A large number of bacterial and fungal species were tested. Specifically, 18 out of 20 of them were clinical isolates (sensitive and resistant to antibiotics). The mechanism was similar to that described above. A 35 min treatment was followed by 2 h of waiting and 10 min of ventilation. Results did not show substantial differences between the various distances, while a significant mean reduction in the microbial load was obtained. However, some significant differences in disinfectant sensitivity were observed between the various microbial species. Strong bactericidal activities, (about 3 log10) were detected for molds and many bacteria (*Pseudomonas* spp., *Enterococcus* spp., *Staphylococcus* spp.). Lower bactericidal activities were observed for the Enterobacteriaceae family (*Enterobacter cloacae, Proteus* spp., *Serratia* spp., and multiresistant *Klebsiella pneumoniae*) and for *Acinetobacter baumannii* (about 1 log10).

A comparative study evaluating the efficacy of 5%, 10%, and 35% HP, showed a 6 log reduction of MRSA after 30 min of 35% HP treatment. The full kill of *Geobacillus stearothermophilus* spores was achieved by using 5% and 10% HP for 70 min [29].

A pilot study investigated the role of HPV in biofilm eradication. An artificial biofilm composed by MDR bacteria (*Acinetobacter baumannii*, *Enterococcus faecalis*, *Klebsiella pneumoniae*, *Pseudomonas aeruginosa*, and *Staphylococcus aureus*) was obtained using a drip-flow reactor. The biofilm was treated with 35% HPV, which allowed a log10 reduction of all the microbial species present in the biofilm in 100 min [30].

In 2014, some authors [31] tested the activity of HPV on various strains of MDR pathogenic microorganisms, including clinical isolates. In particular, seven strains of *Acinetobacter* spp., seven strains of *Klebsiella pneumoniae*, and seven strains of *Pseudomonas aeruginosa* showed minimal inhibitory concentration (MIC) values from 0.5 to 20 mM, after disinfectant exposures from 1 min to 24 h. Similar results were showed on the same microorganisms’ associated biofilms.

Further research groups tested the HPV activity on spore-forming bacteria [32,33,34], such as *Clostridium* spp. [35,36] and *Bacillus* spp. Starting from data obtained from a previous study (Boyce), which highlighted the ability of HPV in spores’ inactivation of a wide range of *Clostridium difficile* strains present on metal disks, the Hospital Saint-Antoine group [37] tested the in vitro activity of 30% HPV. Tests were carried out on the hypervirulent strain of *Clostridium difficile* genotype 027/NAP1/BI using 2 cm^2^ polyvinylchloride (PVC) carriers contaminated with 1 × 10^6^ spores/carriers. Results showed a complete spore eradication after 15 min of 30% HPV treatment.

A similar work was carried out in 2005 by an American group [38], who tested the activity of HPV on different *Bacillus* spp. spores. Variable results were obtained for the different study conditions, such as the different tested surfaces (porous and nonporous).

Several studies showed a good HPV activity against standard strains of tuberculous mycobacteria [39,40,41]. The inactivation of a 5 log inoculum of *Mycobacterium tuberculosis* was achieved in an in vitro test after HPV treatment [42].

Hall et al. [43] tested the HPV antibacterial activity on artificial contamination of *Mycobacterium tuberculosis* H37Rv (ATCC 27294) strain inside biological safety cabinets (BSCs). Results showed a 3 log reduction of *Mycobacterium tuberculosis* in less than 30 min of HPV treatment.

Principal in vitro tests are summarized in Table 1.

## 3. Experimental Test Performed in Hospital Settings with Hydrogen Peroxide Vapor (HPV)

Several studies evaluated the role of HPV in environments contaminated by multidrug-resistant *Acinetobacter baumannii* strains [44,45,46,47] and methicillin-resistant *Staphylococcus aureus* strains [48,49,50].

Experimental tests [51] were directly performed in the hospital setting following the death of 2 out of 13 patients infected by multidrug-resistant *Acinetobacter baumannii.* The same strain was isolated in 7% of the sampled surfaces [52]. After cleaning, the investigated room was subjected to HPV treatment (240 ppm for 8 h). After disinfection, the same microorganism was not detected anymore. The same method, performed on a weekly basis, showed the recolonization of *Acinetobacter baumannii* in some of the tested surfaces after repopulation of the room.

In 2011, further studies highlighted the efficacy of HPV in reducing *Acinetobacter baumannii* complex (ACB) [53] and methicillin-resistant *Staphylococcus aureus* (MRSA) [54] from colonized hospital surfaces after the cleaning and disinfection cycles. The investigated rooms were previously populated by MDR-positive patients. Almost 20 points per room were sampled in over 300 rooms of a 900-bed hospital. Results showed that one standard cleaning/disinfection cycle reduced the only ABC contaminations. HPV treatment (30% for 3–4 h), applied after the end of the cleaning/disinfection cycle, caused a significantly higher efficacy, reducing the number of the rooms colonized by ABC and MRSA. It has been shown that four standard cleaning/disinfection cycles did not eliminate ABC and MRSA contaminations from hospital surfaces, compared to treatment with 30% HPV.

In further studies [55,56], HPV treatment was used to reduce MRSA contamination in the rooms of a 300-bed hospital. After patient discharge, the rooms were cleaned twice. After the introduction of 6% HPV disinfection a reduction of MRSA isolation from environments was obtained. The prevalence of colonized surfaces passed from 25% to 19%.

HPV efficacy on bacteria was also studied by Otter [57], in a study that demonstrated how the application of HPV may be useful in a hospital endemic situation caused by *Acinetobacter* spp. and *Enterobacter cloacae* [58] infections in an intensive care unit (period between June 2005 and March 2006). Transmission of Gram-negative MDR persisted despite the implementation of standard infection-control measures, such as hand hygiene trainings [59,60] and the routine use of sodium hypochlorite for the daily surface cleanings. To disinfect the whole ward and to eradicate the environmental reservoir, a new disinfection procedure was implemented with the use of 30% HPV for 12 h. After the standard cleaning/disinfection procedure, the presence of MDR bacteria was detected on 48% of surfaces. Following the application of HPV, only one MDR bacteria was isolated from the 63 sampled surfaces. No cases of *Acinetobacter* spp. or *Enterobacter cloacae* infections occurred in the four months following the procedure, demonstrating the efficacy of the new disinfection method in an endemic situation. It should be noted that the microorganisms reappeared with pulsed-field gel electrophoresis (PFGE) profiles similar to those present before the new disinfection. This occurrence may originate from the mobile medical equipment transported from outside the wards.

A working group [61] published data related to the efficacy of 30% HPV for the disinfection of 92 intensive care units contaminated by MRSA after patient discharge and routine cleaning. After 1 h and 30 min of HPV disinfection, the authors observed a significant reduction of MRSA counts (*p* = 0.004). After the standard cleaning, 6% of the rooms were still contaminated, while after the HPV treatment this percentage decreased to 0.5%.

A research group [1] assessed the HPV sporicidal activity in hospital rooms previously populated by patients infected by *Clostridium difficile* [62]. Surface samplings carried out before the disinfection showed that 19% of the monitored surfaces were contaminated. After the use of 5% HPV with the addition of silver ions and phosphoric acids for 24 h, a 91% reduction of the contamination was detected (*p* < 0.05). The same test was performed using 0.5% sodium hypochlorite, which reduced only 50% of the contamination.

Another study indicated a significant reduction (*p* < 0.001) in the *Clostridium difficile* infection level at the time when 35% HPV disinfection was implemented. Results showed the reduction in the infection rate from 1.0 to 0.4 cases per 1000 patient-days in the 24 months before HPV usage compared with the first 24 months of HPV usage [63].

Carling [64] compared the disinfection efficacy between a traditional quaternary ammonium compound (QAC) and HPV. Results, obtained by a fluorescent marker test, showed that 40% of surfaces treated with QAC showed a complete removal of bacterial load. HPV treatment led to a sanitization of 77% of surfaces. Since there was no difference in the completeness of the preliminary cleaning (65% and 66%), the significant difference in microbial load reduction is attributable to a better efficacy of HPV which is two times more effective compared to the QAC.

Rutala et al. [65] tested HPV for the decontamination of curtains. Curtains, placed around patient beds and infrequently replaced (every 3–6 months), may represent a reservoir for healthcare pathogens such as MRSA, VRE, *Clostridium difficile*, or other MDR bacteria. In the study, carried out in 37 hospital rooms, the use of HPV reduced the curtains’ microbial load (97%), proving to be a useful method for the decontamination of critical surfaces such as the curtains.

Experimental tests performed in hospital settings with HPV are summarized in Table 2.

## 4. Conclusions

Literature data published by different research groups highlight a good antimicrobial and sporicidal hydrogen peroxide activity. In vitro experiments show that values from 0.3% to 35% of HP may reduce the microbial loads of various opportunistic pathogens (in suspension and on different types of contaminated surfaces). These results encouraged several tests directly performed in the hospital settings. HPV has been used with encouraging results as an additional method associated with the standard disinfection systems. This approach was mostly implemented in wards with severe environmental contamination levels and endemic cases of MDR infections. The application of devices able to atomize hydrogen peroxide in aerosols, at concentrations of not less than 240 ppm and for different contact times, was effective in reducing the contaminations of all the tested microbial species.

Moreover, several encouraging results have been obtained on MDR *Acinetobacter baumannii* and *Clostridium difficile*. In fact, HPV disinfection allowed a 91% reduction of these resistant types of bacterial strains.

Considering the Enterobacteriaceae family, many studies confirmed the in vitro efficacy of HPV against *E. coli*, *Proteus mirabilis*, and *Klebsiella pneumoniae.* A similar efficacy was also observed in hospital wards and intensive care units with surfaces previously colonized by *Enterobacter cloacae*.

Considering the evidence that normal cleaning/disinfection cycles do not guarantee the same efficacy as HPV treatment, it is possible to consider this system as a recommended method for highly contaminated environments. Moreover, this strategy may be useful in situations where routine sanitization and infection-control measures do not stop the infectious transmission cycle.

Encouraging results obtained on Enterobacteriaceae (in vitro also on *Klebsiella pneumoniae*) suggest the possible application of this system in the hospital environment as a new frontier in the prevention and control of healthcare-related infections, mostly in endemic and prolonged infective conditions.

HPV, used in the past as one of the first disinfectants and then forsaken with the introduction of chlorine-based compounds, is therefore proposed today as a highly innovative method for systematic application in the most updated hospital disinfection protocols.

## Figures and Tables

**Table 1 pathogens-09-00408-t001:** Literature data related to in vitro tests evaluating microbial reductions following hydrogen peroxide (HP) or hydrogen peroxide vapor (HPV) treatments at different times and concentrations.

HP or HPV Concentration	Exposure Time	Bacteria Reduction	Author and Year
3000 ppm + 30 ppb silver ions	15 min–24 h	*Escherichia coli* (5 log_10_) *Proteus mirabilis* (6 log_10_) *Klebsiella pneumoniae* (6 log_10_)	Pedahzur, 1995—Davoudi, 2013
50,000 ppm + 0.1% silver ions	5–30 min	*Staphylococcus aureus* ATCC 6538 (8 log_10_) *Pseudomonas aeruginosa* ATCC 15442 (8 log_10_)	De Giglio, 2008
500–600 ppm	2–3 h	*Acinetobacter baumannii* (>5 log_10_) MR *Staphylococcus aureus* (>4 log_10_) MDR *Enterococcus faecalis* (>4 log_10_) *Clostridium difficile* (>2 log_10_)	Lemmen 2015
Glosair™ 400 (hydrogen peroxide with silver ions)	35 min	*Pseudomonas* spp. (>3 log_10_) *Enterococcus* spp. (>3 log_10_) *Staphylococcus aureus* (>4 log_10_) *Acinetobacter baumannii* (>1 log_10_) MDR *Klebsiella pneumoniae—Enterobacter—Proteus* spp. (>1 log_10_)	Herruzo 2014
300,000 ppm	15 min 30 min	*Clostridium difficile* (6 log_10_) *Mycobacterium tuberculosis* (3 log_10_)	Shapey, 2008—Davies, 2011 Hall, 2007
350,000 ppm	30 min 100 min	MR *Staphylococcus aureus* (6 log_10_) Biofilm (*A. baumannii, E. faecalis, K. pneumoniae, P. aeruginosa, S. aureus*) (6 log_10_)	Murdoch 2016 Watson 2018

MR: Methicillin-resistant; MDR: multidrug-resistant bacteria.

**Table 2 pathogens-09-00408-t002:** Literature data related to experimental tests performed in hospital settings. Percentage of contamination following hydrogen peroxide vapor (HPV) treatments at different times and concentrations.

HPV Concentration	Exposure Time	Contamination Reduction	Author and Year
240 ppm	8 h	MDR *Acinetobacter baumannii* (from 7% to 0%)	Ray, 2000 Saed, 2006
50,000–60,000 ppm	3–4 h 24 h	MR *Staphylococcus aureus* (from 25% to 19%) *Clostridium difficile* (from 19% to 0.2%)	Dancer, 2008 Mitchell, 2014 Barbout, 2009–2012 Shaughnessy, 2011
300,000 ppm	1 h 12 h	MR *Staphylococcus aureus* (from 100 to 40 CFU/cm^2^) MDR *Acinetobacter baumannii—Enterobacter cloacae* (from 48% to 2%)	Bartels, 2008 Blazejewski, 2015 Lacey, 1995

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
