# Peer review of "Role of Hydrogen Peroxide Vapor (HPV) for the Disinfection of Hospital Surfaces Contaminated by Multiresistant Bacteria"

_pathogens, 2020, doi:10.3390/pathogens9050408_

Round 1
Reviewer 1 Report
I only have one minor suggestion:
Tables 1 and 2, when talking about the concentrations of hydrogen peroxide, percentage and ppm were used. It would be better to use molarity for all the studies so that readers can compare the concentration of hydrogen peroxide used in different studies.
Author Response
Tables 1 and 2, when talking about the concentrations of hydrogen peroxide, percentage and ppm were used. It would be better to use molarity for all the studies so that readers can compare the concentration of hydrogen peroxide used in different studies.
Dear reviewer, thank you for your comment. Tables have been modified as requested.
Reviewer 2 Report
In the medical literature we can find several data concerning germicidal effectiveness of hydrogen peroxide in health -care settings. Hydrogen peroxide is active against a wide range of microorganisms. The presented literature overview was needed because of looking for good protection measures against microorganisms in the hospitals. The authors analyze the usefulness of hydrogen peroxide also in the form of vapour.
The whole was written easily and takes into account the most important aspects of this disinfectant.
Author Response
In the medical literature we can find several data concerning germicidal effectiveness of hydrogen peroxide in health -care settings. Hydrogen peroxide is active against a wide range of microorganisms. The presented literature overview was needed because of looking for good protection measures against microorganisms in the hospitals. The authors analyze the usefulness of hydrogen peroxide also in the form of vapour.
The whole was written easily and takes into account the most important aspects of this disinfectant.
Dear reviewer, thank you for your attention and comments.